# Resonant thermal energy transfer to magnons in a ferromagnetic nanolayer

Michal Kobecki [1✉], Alexey V. Scherbakov[1,2✉], Tetiana L. Linnik [3], Serhii M. Kukhtaruk [1,3], Vitalyi E. Gusev [4], Debi P. Pattnaik[5], Ilya A. Akimov [1,2], Andrew W. Rushforth [5], Andrey V. Akimov [5] & Manfred Bayer [1,2]

Energy harvesting is a concept which makes dissipated heat useful by transferring thermal energy to other excitations. Most of the existing principles are realized in systems which are heated continuously. We present the concept of high-frequency energy harvesting where the dissipated heat in a sample excites resonant magnons in a thin ferromagnetic metal layer. The sample is excited by femtosecond laser pulses with a repetition rate of 10 GHz, which results in temperature modulation at the same frequency with amplitude ~0.1 K. The alternating temperature excites magnons in the ferromagnetic nanolayer which are detected by measuring the net magnetization precession. When the magnon frequency is brought onto resonance with the optical excitation, a 12-fold increase of the amplitude of precession indicates efficient resonant heat transfer from the lattice to coherent magnons. The demonstrated principle may be used for energy harvesting in various nanodevices operating at GHz and sub-THz frequency ranges.

[1] Experimentelle Physik 2, Technische Universität Dortmund, Otto-Hahn-Strasse 4a, 44227 Dortmund, Germany. [2] Ioffe Institute, Politechnycheskaya 26, St. Petersburg, Russian Federation 194021. [3] Department of Theoretical Physics, V.E. Lashkaryov Institute of Semiconductor Physics, Pr. Nauky 41, Kyiv 03028, Ukraine. [4] LAUM, CNRS UMR 6613, Le Mans Université, 72085 Le Mans, France. [5] School of Physics and Astronomy, University of Nottingham, Nottingham NG7 2RD, UK. ✉email: michal.kobecki@tu-dortmund.de; alexey.shcherbakov@tu-dortmund.de

The transfer of thermal energy to mechanical, electrical, or magnetic excitations is of great interest and is widely considered for energy harvesting when waste heat is transferred to more usable types of energy[1]. The rational utilization of heat is a critical task for nanoscale electronics, where operations are accompanied by extensive production of parasitic heat. Nanoscale devices for communication and computing operate with digital signals, which are generated by ultrafast current or optical pulses with high repetition rate. The heat generated in this case is also modulated at the same frequency and could be transferred to nonthermal types of excitation, which become the elements of the energy-harvesting process. This process would be most efficient in the case of resonance, when the modulated heat is transferred to a subsystem with the same intrinsic frequency. Resonant heat transfer is widely used in photoacoustics and was proposed by Bell in 1881[2]. There the modulated optical signal is absorbed in a system with acoustic resonance at the frequency of modulation and as a result the modulated heat resonantly excites the acoustic wave. Modulation frequencies in photoacoustics do not exceed the MHz frequency range[3], while thermomodulation with frequencies up to 400 GHz[4] is used in picosecond ultrasonics for studying sub-THz coherent phonon dynamics.

Here we propose to convert high-frequency (GHz) modulated heat to magnons, which are collective spin excitations in magnetically ordered materials. The manipulation of coherent high-frequency magnons on the nanoscale is one of the most prospective concepts for information technologies[5], also in the quantum regime[6]. While the most common way to excite coherent magnons is nonthermal and based on microwave techniques[7], thermal methods have proven successful in ferromagnetic metals. These methods are based on ultrafast modulation of the magnetic anisotropy induced by rapid lattice heating, for example, due to the absorption of optical pulses[8,9]. In ferromagnetic metals, the intrinsic magnon frequency depends on the external magnetic field and may be varied between ~1 and ~100 GHz[10]. This range covers the clocking frequency for most electronic devices and, thus, magnons are suitable to receive dissipated heat modulated at the resonant frequency.

In the present paper, we introduce the concept of transferring heat dissipated in a semiconductor from the relaxation of hot electrons excited by a 10 GHz laser to coherent magnons. In our experiments, the energy harvester is a metallic ferromagnetic film of 5 nm thickness grown on a semiconductor substrate. We demonstrate an efficient heat–magnon transfer by measuring a 12-fold increase of the fundamental magnon mode amplitude at the resonance conditions. The experiments and related theoretical analysis show that GHz modulation of the temperature on the scale of ~0.1 K is sufficient for the excitation of magnons with amplitude reliable for the operation of spintronic devices.

## Results

### Samples and experimental technique.
The ferromagnetic energy harvester is a 5 nm layer of Galfenol ($Fe_{0.81}Ga_{0.19}$) grown by magnetron sputtering on a (001) semi-insulating GaAs substrate and covered by a 2 nm Cr cap layer to prevent oxidation. The chosen composition of Fe and Ga is characterized by a high Curie temperature ($T_c \approx 900$ K) and large saturation magnetization $M_0$[11]. Experiments were carried out at ambient conditions at room temperature with an external magnetic field, **B**, applied in the layer plane at an angle $-\pi/8$ from the [100] crystallographic direction, which corresponds to the maximal sensitivity of the magnetization, **M**, to the temperature-induced changes of the magnetic anisotropy[12]. In the studied layer, the lowest, fundamental mode of the quantized magnon spectrum is well separated from the higher-

order modes due to the large exchange mode splitting[12]. As a result, the magnon spectrum consists of a narrow single spectral line of Lorentzian shape with a width of ≈500 MHz. Examples of the magnon spectrum for several values of $B$ measured by monitoring the magnetization precession excited by a single optical pulse are shown in Fig. 1a (for details see "Methods" section). The experimentally measured dependence of magnon frequency $f$ on $B$ is shown by the symbols in the right panel of Fig. 1a.

Figure 1b illustrates a schematic of the pump–probe experiment for magnon energy harvesting from high-frequency modulated heat. A femtosecond pump laser with repetition rate $f_0 = 10$ GHz and average power up to $W = 150$ mW is used for modulation of the lattice temperature. A second laser with repetition rate of 1 GHz and average power of 18 mW is used to probe the response of the magnons to the high-frequency heat modulation by means of the transient polar Kerr rotation (KR) effect. The pump and probe beams are focused on the $Fe_{81}Ga_{19}$ layer into overlapping spots of 17 and 14 μm diameter, respectively, using different sectors of the same reflective microscope objective (for details see the "Methods" section).

### Temperature evolution.
Figure 1c (upper panel) shows the calculated temporal evolution of the lattice temperature $T(t)$ in the $Fe_{0.81}Ga_{0.19}$ layer under 10 GHz pump optical excitation. The calculations were performed by solving the heat equations taking into account the thermal resistance at the $Fe_{0.81}Ga_{0.19}$/GaAs interface (see "Methods" section and Supplementary Notes 1 and 2). The temperature modulation amplitude $\delta T \sim W/(C_f r^2)$, where $C_f$ and $r$ are the heat capacity and radius of the excitation spot. It is seen that the lattice temperature oscillates with amplitude $\delta T$ on a stationary background that exceeds the room temperature $T_0$ by $\Delta T$. Both $\Delta T$ and $\delta T$ increase linearly with $W$: $\delta T = pW$ and $\Delta T = PW$, where $p$ and $P$ are constants, which for our structure are equal to 4.9 and 510 K W$^{-1}$, respectively. The temperature modulation is not harmonic and its spectrum, which is shown in the lower panel of Fig. 1c, consists of discreet harmonics at the frequencies $f_n = nf_0$, where $n$ is an integer. The harmonic amplitude decreases with the increase of $n$.

The idea of the experiments illustrated in Fig. 1a is to exploit the periodic thermal modulation for exciting coherent magnons and to monitor the magnetization precession by transient KR measurements of the out-of-plane magnetization projection, $\delta M_z$, for various values of $B$. We expect a resonant increase of the precession amplitude at the resonances when $B = B_n$, which correspond to $f = f_n$. The expected values of $B_n$ for the first three harmonics are shown in the right panel of Fig. 1a by the vertical arrows. The idea is analogous to the excitation of a harmonic oscillator with tunable eigenfrequency $f$ by a periodic force, which acts on the oscillator with repetition rate $f_0$. For magnetization precession, this force is generated by the modulated temperature $T(t)$, and the magnon eigenfrequency $f$ is controlled by the external magnetic field.

### Resonant magnon signals and their spectra.
Figure 2 shows the transient KR signals measured for $W = 95$ mW and various values of $B$. The signals measured in the vicinity of the first resonance ($n = 1$) have a harmonic shape. At these conditions, we fit the signals by a sine function $\delta M_z(t) = A_B \sin(\omega_1 t + \varphi_B)$, where $\omega_1 = 2\pi f_1$, and $A_B$ and $\varphi_B$ are the magnetic field-dependent amplitude and phase of the harmonic oscillations. Times $t = 0$ and 100 ps correspond to excitation of the sample by the pump pulses. It is seen that the amplitude of the signal is maximal when $B = B_1 = 44$ mT, which corresponds to the first resonance $f = f_1$ (see Fig. 1a). In the vicinity of the resonance, the phase changes from $\varphi_B = -\pi/2$ for $B < B_1$ to $\varphi_B = +\pi/2$ for $B > B_1$. At the

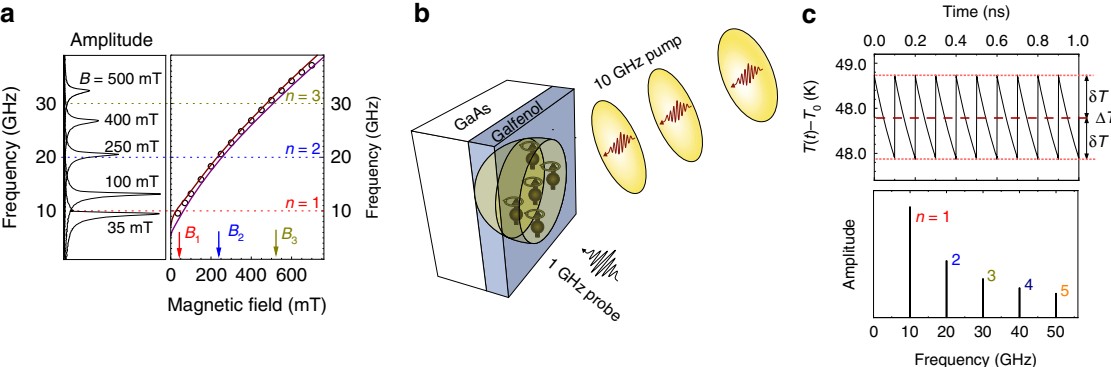

**Fig. 1 Magnon resonances and temperature modulation. a** Fast Fourier transform spectra showing the fundamental magnon mode for several values of magnetic field, $B$ (left panel) and the field dependence of its frequency, $f$ (right panel) for the 5-nm-thick $Fe_{0.81}Ga_{0.19}$ energy-harvesting layer. Symbols show $f(B)$ obtained by fast Fourier transformation of Kerr rotation signals measured in a single pulse pump–probe experiment. Solid lines are calculated dependences for $\Delta T = 0$ K (upper curve) and $\Delta T = 200$ K (lower curve). The dashed horizontal lines show the frequencies of the harmonics in the temperature modulation spectrum induced by the 10 GHz optical excitation. The vertical arrows indicate the expected resonances for the fundamental magnon mode at these harmonics. **b** Scheme of the experiment. **c** Calculated temporal evolution of the $Fe_{0.81}Ga_{0.19}$ lattice temperature induced by the 10 GHz optical excitation of the $Fe_{0.81}Ga_{0.19}$/GaAs heterostructure at the pump excitation power $W = 95$ mW (upper panel) and its Fourier spectrum (lower panel).

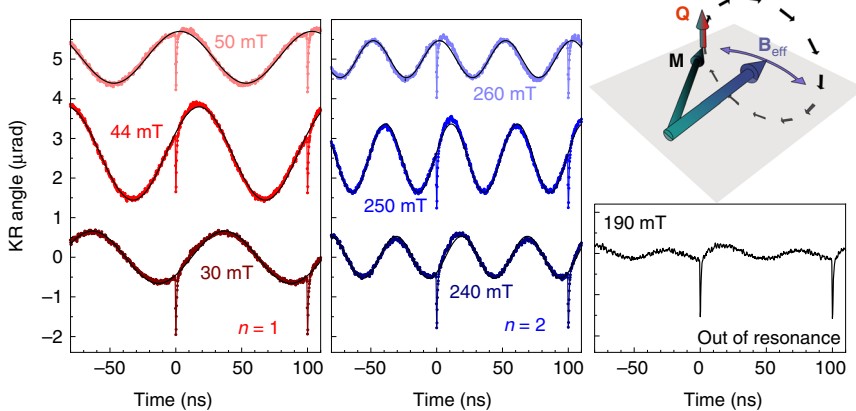

**Fig. 2 Magnon signals in time domain.** Transient Kerr rotation signals measured in the vicinity of the first (left panel) and second (central panel) resonances and at the intermediate magnetic field of 190 mT, corresponding to off-resonance (right panel). The amplification of the precession amplitude observed at $B = B_1 = 44$ mT and $B = B_2 = 250$ mT is due to thermally induced resonant driving of the precession **M** by the oscillating effective field $\mathbf{B}_{eff}$. The torque **Q** acting on the magnetization (see sketch at the top right) is maximal at the resonance conditions: $f = nf_0$.

resonance $\varphi_B \approx 0$. Similar results are observed for the signals measured in the vicinity of the second resonance ($n = 2$). The amplitude of the oscillations at $B = B_2 = 250$ mT are 1.4 times smaller than for the first resonance and the same conclusions as for the first resonance can be made concerning the field dependences of the amplitude and phase of the signal around $B_2$. Away from the resonances the signals are periodic but not harmonic. An example of a nonresonant signal measured at $B = 190$ mT is presented in the right panel of Fig. 2. The amplitude of the signal is much smaller than in the case of resonance and the temporal shape is impossible to fit with a single sine function.

To plot the dependence of the measured signal amplitude as a function of $B$, we present the root-mean-square (RMS) amplitude $\tilde{A}_B$ of the measured KR signal. The symbols in Fig. 3a show the field dependences $\tilde{A}_B(B)$ for three pump excitation powers $W$. It is clearly seen that the dependences have peaks at the resonant values of $B_n$ corresponding to $n = 1$, 2, and 3. Figure 3b shows zoomed fragments of the field dependences of the amplitude and phase, respectively, obtained for the first resonance at $W = 95$ mW. The 12-fold increase of $\tilde{A}_B$ at the resonance condition is clearly seen by comparison with the out-of-resonance RMS

amplitude shown by the horizontal dashed line in the upper panel of Fig. 3b. No peaks in the dependence of the precession amplitude on $B$ are detected in the case of single pulse excitation, where $\tilde{A}_B$ gradually decreases with the increase of $B$.

The values of the measured resonance fields $B_n$ shift to slightly higher fields when $W$ increases. We explain this shift by the heat-induced decrease of the magnon frequencies[13] and respective increase of the magnetic field value required for achieving the resonance conditions. We use the values of this shift to obtain the background temperature $\Delta T$ of the Galfenol film by comparison with the known dependence of $f(B)$ on temperature. Two dependences of $f(B)$ calculated using the known dependence of the Galfenol magnetic parameters on temperature[13,14] are demonstrated in the right panel of Fig. 1a by the solid lines (for details see Supplementary Note 2). The corresponding values of the background temperature obtained from the experimentally measured shifts of the resonances are $\Delta T = 28$, 44, and 77 K for $W = 35$, 55, and 95 mW, respectively. They are 40% higher than the values calculated theoretically from the heat equations. We attribute this difference to the additional background heating by the probe beam that is not considered in the theoretical modeling.

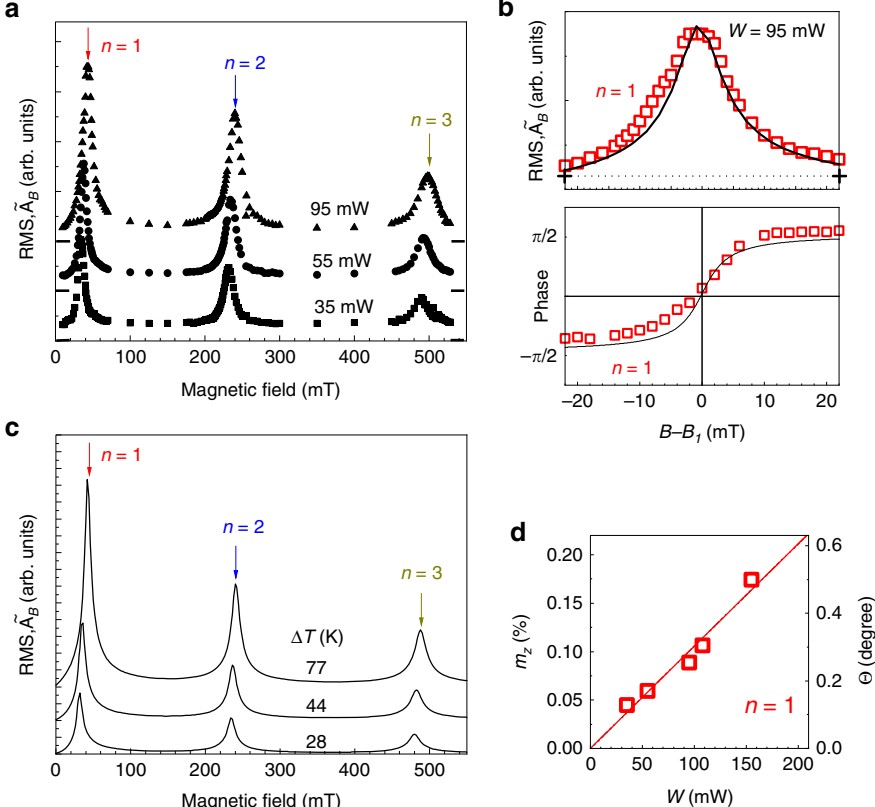

**Fig. 3 Magnon spectra. a** Measured magnetic field dependences of the root-mean-square (RMS) amplitude of the Kerr rotation signal, $\tilde{A}_B$, for three values of pump excitation power, $W$. The vertical arrows indicate the magnetic fields $B_n$ at which the resonance condition $f = f_n$ is fulfilled; the horizontal bars indicate zero signal levels. **b** Measured (symbols) and calculated (solid lines) zoomed fragments of the field dependences of $\tilde{A}_B$ (upper panel) and the phase (lower panel) in the vicinity of the first resonance ($n = 1$) for $W = 95$ mW; the dashed horizontal line shows $\tilde{A}_B$ at the intermediate off-resonance field. **c** Calculated field dependences of the RMS precession amplitude for the background temperatures $\Delta T$ obtained from the experimental dependences in **a**. **d** Power dependence of the relative precession amplitude (left axis) and the corresponding precession angle (right axis) at the first resonance ($n = 1$) when $f = f_0$.

## Discussion

In the analysis, we consider thermal modulation of the magnetic anisotropy as the main mechanism for magnon excitation in our experiment and do not take into account other mechanisms (e.g., thermal strain). This approach is based on previous experiments and theoretical analysis where various mechanisms of laser-pulse-induced excitation of magnons in Galfenol were considered[14]. We also exclude the effect of thermal gradients inside a ferromagnetic film (we estimate 4% temperature difference across the film). This gradient in special cases can induce alternating current (a.c.) spin transfer[15,16] modulated at the laser repetition rate, $f_0$. In our experiment, due to the uniform in-plane magnetization in a single thin ferromagnetic layer, this transfer does not produce a torque on the magnetization[17]. The dominating role of the thermal modulation of the magnetocrystalline anisotropy is also confirmed by the dependence of the precession amplitude excited by a single laser pulse on the direction of the external magnetic field. This dependence demonstrates a four-fold in-plane symmetry with a slight uniaxial distortion, which corresponds to the magnetocrystalline anisotropy of the studied layer[12].

We describe the effect of thermal modulation on the magnetization, **M**, by considering the magnetization precession motion in a time-dependent effective field $\mathbf{B}_{\text{eff}}$ as schematically shown in Fig. 2[14]. The magnetization precession is described by the Landau–Lifshitz–Gilbert (LLG) equation[7]:

$$\frac{d\mathbf{m}}{dt} = -\gamma_0 \mathbf{m} \times \mathbf{B}_{\text{eff}}(t) + \alpha_0 \mathbf{m} \times \frac{d\mathbf{m}}{dt}, \quad (1)$$

where $\mathbf{m} = \mathbf{M}/M_0$ is the normalized magnetization and $\alpha_0$ is the Gilbert damping parameter. The effective magnetic field is determined as $\mathbf{B}_{\text{eff}} = -\nabla_{\mathbf{m}} F_{\mathbf{M}}(\mathbf{m}, t)$, where $F_{\mathbf{M}}$ is the normalized free energy density[7]:

$$F_{\mathbf{M}}(m) = -(\mathbf{m} \cdot \mathbf{B}) + B_{\text{d}} m_z^2 + K_1 \left( m_x^2 m_y^2 + m_z^2 m_y^2 + m_x^2 m_z^2 \right) - K_{\text{u}}(\mathbf{m} \cdot \mathbf{s})^2, \quad (2)$$

where the first term is the Zeeman energy, the second term is the demagnetization energy $\left( B_{\text{d}} = \frac{\mu_0 M_0}{2} \right)$, and the following two terms describe the cubic and uniaxial magnetocrystalline anisotropy with respective coefficients $K_1$ and $K_{\text{u}}$, with the unit vector $\mathbf{s} \parallel [110]$ along the uniaxial anisotropy axis. In the chosen coordinate system, the $x$, $y$, and $z$ axes correspond to the main crystallographic directions [100], [010], and [001] (normal to the layer plane), respectively. At equilibrium, that is, without temperature modulation, the value of $\mathbf{B}_{\text{eff}}$ determines the fundamental magnon frequency, while its direction sets the equilibrium orientation of the magnetization **m**. The change of temperature alters $\mathbf{B}_{\text{eff}}$ through the temperature-dependent parameters $M_0$, $K_1$, and $K_{\text{u}}$. Within the studied temperature range their dependence on $T$ is linear[13]: $X = X^{\text{RT}} + \beta_X(T - T_0)$, where $X = M_0$, $K_1$, or $K_{\text{u}}$, the index RT indicates the room-temperature values and $\beta_X$ is the corresponding thermal coefficient. The room-temperature values $M_0^{\text{RT}} = 1.96$ T, $K_1^{\text{RT}} = 20$ mT, and $K_{\text{u}}^{\text{RT}} = 9$ mT for our sample are found from fitting the experimental dependence $f(B)$ (shown in the right panel of Fig. 1a)[12,14]. The thermal

coefficients, $\beta_{M_0} = -0.97\,\text{mT K}^{-1}$, $\beta_{K_1} = -0.046\,\text{mT K}^{-1}$, and $\beta_{K_u} = -0.025\,\text{mT K}^{-1}$ for Galfenol are taken from previous studies[13,14]. The Gilbert damping coefficient $\alpha_0 = 0.006$ is obtained from the precession kinetics excited by a single laser pulse.

The modulation amplitude $\delta T << \Delta T$ and for solving Eq. (1), we assume that the magnitude of $\mathbf{B}_{\text{eff}}$ is constant and the temperature modulation affects only its direction. We also consider only the ground magnon mode with zero wave vector, that is, we assume a uniform precession, due to the large diameter (17 μm) of the excitation spot, which limits the in-plane wave vector of magnons to $q_{||} \leq 4700\,\text{cm}^{-1}$. In the 5-nm-thick Galfenol layer, the frequency range for such $q_{||}$ does not exceed 270 MHz[18,19], which is clearly below the 500 MHz spectral width of the fundamental magnon mode. For details of the magnon dispersion and contribution of magnons with finite wave vectors in a 100-nm-thick layer, see Supplementary Note 4.

With these assumptions, the LLG equation can be reduced to a system of equations (see the "Methods" section for details) for the magnetization components, where the thermal modulation of the anisotropy coefficients alters the direction of $\mathbf{B}_{\text{eff}}$ and generates a tangential driving torque, $\mathbf{Q}$, acting on the magnetization[14] as schematically shown in the inset of Fig. 2. The vector $\mathbf{B}_{\text{eff}}$ follows the temporal temperature evolution $T(t)$ shown in Fig. 1c, which can be written as sum of background heating and a Fourier series expansion for the small periodic temperature modulation:

$$T(t) = T_0 + \Delta T + \delta T \sum_{n=1}^{\infty} u_n \sin(\omega_n t + \psi_n). \qquad (3)$$

Here the amplitude and phase of the $n$th harmonic are $u_n = \frac{2}{\pi}\frac{\omega_1 \tau_0}{\sqrt{1+\omega_n^2 \tau_0^2}}$ and $\psi_n = \arctan(\frac{1}{\omega_n \tau_0})$, respectively, $\omega_n = 2\pi f_n$, $\tau_0$ is the characteristic cooling time of the Galfenol layer, which is determined by the thermal boundary resistance, $R$, at the interface with GaAs[20]. The solution of the linearized Eq. (1) for the $\delta m_z$ component expanded as a Fourier series has the form:

$$\delta m_z(t) = \sum_{n=1}^{\infty} A_n(B)\sin(\omega_n t + \varphi_n(B)), \qquad (4)$$

where $A_n(B)$ and $\varphi_n(B)$ are the time-independent amplitude and phase for the $n$th harmonic. Like in the case of a simple harmonic oscillator, at the resonant conditions $f(B) = f_n$ the precession amplitude reaches maximum with zero phase-shift relative to the driving force $\varphi_n(B) \approx 0$. The analytical expressions for $A_n(B)$ and $\varphi_n(B)$ are presented in the "Methods" section.

Figure 3c shows the field dependences of the RMS amplitude of the calculated $\delta m_z(t)$ for three values of the background temperature $\Delta T$, which correspond to the $W$ indicated near the experimentally measured data in Fig. 3a. The agreement of the measured and calculated spectral shapes is explicitly demonstrated in the zoomed dependences of the RMS amplitude and phase in Fig. 3b, where the calculated dependences (solid lines) for $n = 1$ are presented together with the experimental values. Good agreement is observed also for higher $n$ and $W$.

The measured power dependence of the resonant magnon amplitude for $n = 1$ in the studied sample is shown in Fig. 3d. The precession amplitudes are obtained from the absolute values of the KR angle in the transient KR signals[12]. The field-independent small amplitude background, which is due to the response of the electron system and not related to the magnetization[21], has been subtracted from the experimental values. At $W = 100\,\text{mW}$ the precession amplitude reaches 0.1% of the saturation magnetization $M_0$. To get this value, the parameters of the magnetic anisotropy have to be modulated by the oscillating temperature with the amplitude $\delta T = 0.54\,\text{K}$. This is in perfect agreement with the

values obtained by solving the thermal equations, which for the same power give $\delta T = 0.49\,\text{K}$. The achieved precession amplitude, which corresponds to the precession angle $\Theta \approx 0.25°$ (with ellipticity of 0.2) and the generated a.c. induction $\Delta M_z \sim 1\,\text{mT}$, is large enough for prospective spintronics applications: detection of spin currents by spin pumping[22,23]; manipulating single spin states of NV centers in nanodiamonds by microwave magnetic fields[24]; excitation of propagating spin waves in magnonic devices[25].

For the practical use of resonant energy harvesting in a device excited optically or electrically, it is important to achieve high efficiency of the transformation of the modulated part of the thermal energy into the oscillations of magnetization. This efficiency is defined by the ratio of the magnetization angle $\Theta$ of precession and the power $W$ injected into a device. In our experiment, we demonstrate $\Theta/W = 2.5\,°\text{W}^{-1}$, which is only one order of magnitude less than the efficiency of conventional mesoscopic microwave devices[26] and of the same order as devices utilizing surface acoustic waves for spin pumping[23]. At resonance, the magnetization precession remains harmonic up to the maximum used power, and, thus, nonlinear (anharmonic) effects do not affect the transformation efficiency. However, the shift of the resonance frequency with the increasing $\Delta T$ at higher $W$ affects the linear power dependence of the precession amplitude at fixed magnetic field. This temperature-induced "nonlinearity" is known from the conventional microwave experiments on ferromagnetic resonance[27]. The effect of this nonlinearity is very weak in the studied structure. For instance, for the second resonance ($n = 2$) the 20% change of the background temperature without changing thermal modulation amplitude results in a ~3% decrease of the precession amplitude. Thus, stability of the background temperature is not critical for the suggested concept.

The dependences of $\delta T$ on the parameters of the used structure and materials define the efficiency of the heat transfer to the magnons. For optical excitation, shorter penetration depths and smaller heat capacities of the ferromagnetic layer increase $\delta T$ and correspondingly $\Theta/W$. It is convenient to define the dimensionless resonant harvesting efficiency, $\zeta = \delta T/\Delta T$, which governs the ratio of the useful temperature modulation $\delta T$ relative to the parasitic heating $\Delta T$. Obviously, it is favorable to have $\zeta$ as large as possible. For instance, faster cooling to the bulk of the substrate leads to an increase of $\delta T$ with a simultaneous decrease of $\Delta T$ and correspondingly to an increase of $\zeta$. In our particular experiment, where both the $Fe_{0.81}Ga_{0.19}$ nanolayer and the GaAs substrate are excited optically, we get $\zeta \sim 10^{-2}$. The value of $\zeta$ increases with the decrease of the thickness of the ferromagnetic layer and the increase of the substrate thermoconductivity (see Supplementary Note 3). For instance, for the same ferromagnetic layer on a silicon substrate $\zeta$ increases by a factor of 1.5. An efficient way to increase $\zeta$ and the efficiency of the heat–magnon transfer would be to use a multilayer structure where $\Delta T$ is governed by the rapid escape of heat to the substrate while the high-frequency thermal waves, which define $\delta T$, are localized inside or in the vicinity of the harvester layer. To demonstrate that the concept is applicable for electrical nanodevices, we have considered the case of a thin conducting layer (e.g., graphene) where Joule heat can be generated as a result of passing GHz current pulses and transferred to the ferromagnetic harvester through a thin spacer layer. For high efficiency, the spacer layer should have high thermal conductivity, while the ferromagnetic layer should have low specific heat capacity. In the example presented in the Supplementary Note 3, we get $\zeta = 4 \times 10^{-3}$.

In our experiments, we excite resonances up to 30 GHz. To reach higher, sub-THz frequencies, the corresponding bandwidth should be available for both temperature and magnon modulation. For temperature modulation, we refer to the picosecond acoustic experiment[4] where the temperature in a 12-nm Al film was modulated at a frequency up to 400 GHz. The thermal

properties of ferromagnetic metals do not differ much from normal metals and thus similar modulation in thin films is achievable. For optical excitation of metals, it is essential that optically excited hot electrons pass their energy to the lattice in a time <1 ps. Theoretically, for the parameters fixed as above, the amplitude of the temperature oscillations is proportional to the laser power and inversely proportional to the frequency up to sub-THz frequencies. For the magnons in thin (Fe,Ga) films, narrow-band precession with frequencies up to 100 GHz was demonstrated at strong magnetic fields[12]. Moreover, metallic ferrimagnetic materials (e.g., Mn-Te compounds) possess narrow magnon resonances at frequencies up to hundreds of GHz at low magnetic fields[28]. At the lower frequency end, there is no limit for temperature modulation, while for magnons the lower frequency varies from hundreds of MHz (e.g., for garnets) up to several GHz in metallic ferromagnets. The available choice of a proper magnetic material is not limited to the Galfenol used in our study. This ferromagnet is actively studied nowadays[29] and presents a good example for realization of the demonstrated concept due to its pronounced magnetocrystalline anisotropy and narrow magnon resonances (long lifetime of precession). However, other ferro- and ferrimagnetic materials possessing similar properties will also work as resonant harvesters for transferring parasitic heat modulated at high frequencies to magnons.

To conclude, we have demonstrated how the heat generated during 10 GHz pulsed optical excitation is used for the excitation of resonant magnons. The amplitude of the magnetization precession at the fundamental magnon frequency increases enormously when the repetition rate of the optical pulses is equal to the magnon frequency. The resonances also occur at the higher harmonic frequencies, that is, 20 and 30 GHz. An amplitude of the temperature modulation of order ~0.1 K is sufficient to generate a.c. magnetization with an amplitude of 1 mT[30], which is sufficient to be exploited in various applications including information and quantum technologies. The concept of resonant heat transfer can be considered as a prospective method to generate magnons in ferromagnetic layers deposited on processors and other microchips operating at GHz and sub-THz clock frequencies.

## Methods

**Pump–probe measurements with 10 GHz repetition rate.** We used two synchronized Titanium sapphire lasers (Taccor x10 and Gigajet 20c from Laser Quantum) generating 50 fs pulses with repetition rates of 10 GHz (pump) and 1 GHz (probe) at center wavelengths of 810 and 780 nm, respectively. The beams were focused onto the sample using a reflective microscope objective with a magnification factor of 15 comprising 4 sectors through which light could enter and exit the objective[31]. The incidence angles of the laser beams are 17°. The spot diameters for the pump and probe beams were set to 17 and 14 μm, respectively. The diameters are defined at the $1/e^2$ level of the Gaussian spatial energy distributions. The coherent response of the magnetization was measured by monitoring the polar KR of the reflected probe pulses using a differential scheme based on a Wollaston prism and a balanced optical receiver with 10 MHz bandwidth. Temporal resolution was achieved by means of an asynchronous optical sampling (ASOPS) technique[32], where we used an offset frequency of 20 kHz between the 10 GHz laser and the tenth harmonics of the 1 GHz laser in order to resolve the transient signals in 100 ps time window between pump pulses with the time resolution of 50 fs. The sample was mounted between the poles of a dipole electromagnet.

**Measurements of the magnon spectrum with single pulse excitation.** The magnon spectrum of the studied sample was obtained by a conventional magneto-optical pump–probe scheme based on two mode-locked Erbium-doped ring fiber lasers (FemtoFiber Ultra 780 and FemtoFiber Ultra 1050 from Toptica). The lasers generate pulses of 150 fs duration with a repetition rate of 80 MHz at wavelengths of 1046 nm (pump pulses) and 780 nm (probe pulses). The magnetization precession was excited by the pump pulses with an energy of 3 nJ per pulse focused on the Galfenol film surface to a spot of 17 μm diameter. The linearly polarized probe pulses with energy of 30 pJ per pulse hitting at normal incidence to the sample surface were focused to a spot of 1 μm diameter in the center of the pump spot. The coherent response was measured by transient polar KR in the same way as for the 10 GHz measurements. Temporal resolution was achieved also by the ASOPS

technique with a frequency offset of 800 Hz, which in combination with the 80 MHz repetition rate allowed measurement of the time-resolved signals in the time window of 12.5 ns with 150 fs time resolution. The magnon spectra and the magnetic field dependence of the central frequency were obtained by fast Fourier transforming the corresponding transient KR signals.

**Modeling of temperature evolution.** In the case of periodic laser-pulse excitation, the temperature oscillates around the background temperature $T_0 + \Delta T$ with amplitude $\delta T$. The amplitude $\delta T$ is estimated by solving the standard heat equations for the lattice temperatures of $Fe_{0.81}Ga_{0.19}$ and GaAs for periodic excitation by laser pulses using Comsol Multiphysics®[33] (see Supplementary Note 1). We estimate the background heating $\Delta T$ from the solution of the 3D stationary heat equation for continuous wave laser excitation (see Supplementary Note 2). We use the following parameters for GaAs[34]: refractive index $3.7 + i0.09$, which results in an absorption length of 740 nm; heat capacity $1.76 \times 10^6$ J m$^{-3}$ K$^{-1}$ and thermal conductivity 55 W m$^{-1}$ K$^{-1}$[35,36]. We assume that the optical and thermal parameters of $Fe_{0.81}Ga_{0.19}$ are close to the parameters of Fe: refractive index $2.9 + i3.4$[37]; heat capacity $3.8 \times 10^6$ J m$^{-3}$ K$^{-1}$ and thermal conductivity 80 W m$^{-1}$ K$^{-1}$[38]. As a result the reflection coefficient of the (Fe,Ga)/GaAs heterostructure is 0.4 and the absorption coefficients for the $Fe_{0.81}Ga_{0.19}$ layer and GaAs substrate are 0.1 and 0.49, respectively. The calculated values of $\Delta T$ and $\delta T$ at the excitation power $W = 95$ mW are 48 and 0.47 K, respectively.

In our calculations, we consider the thermal boundary resistance, $R$, which determines the flux through the $Fe_{0.81}Ga_{0.19}$/GaAs interface[20]. In order to estimate $R$, we measured the reflectivity signal from the studied structure in the case of low-frequency (80 MHz) laser excitation. Using the experimentally observed decay time of the transient reflectivity $\tau_0 = 200$ ps, we found the thermal boundary resistance $R = 10^{-8}$ m$^2$ K W$^{-1}$.

**Modeling the periodic laser-pulse-induced magnetization precession.** To calculate the magnetization precession amplitudes $A_n(B)$ and phases $\varphi_n(B)$, we apply the approach developed in ref. [14] where the detailed theory of magnetization precession due to ultrafast heating of a Galfenol film was considered. We analyze the precession by rewriting the LLG equation in a spherical coordinate system with in-plane azimuthal angle, $\phi$, and polar angle, $\theta$ ($\theta = 0$ corresponds to the layer normal, $z$), which describes the direction of the normalized magnetization, $\mathbf{m}$. Assuming that the changes of $\delta\phi$ and $\delta\theta$ from the equilibrium angles $\phi_0$ and $\theta_0$ are small and induced by the modulation of the cubic and uniaxial anisotropy constants, the LLG equation in linear approximation has the form:

$$\frac{\partial \delta\theta}{\partial t} = \gamma_0 F_{\theta\theta}\delta\theta + \alpha_0 \frac{\partial \delta\theta}{\partial t},$$
$$\frac{\partial \delta\theta}{\partial t} = \gamma_0 F_{\phi\phi}\delta\phi - \gamma_0 F_{\phi K_1}\delta K_1(t) - \gamma_0 F_{\phi K_u}\delta K_u(t) - \alpha_0 \frac{\partial \delta\phi}{\partial t},$$

(5)

where the $F_{i,j} = \frac{\partial^2}{\partial i \partial j} F_{\mathbf{M}}(i,j = \theta, \phi, K_1, K_u)$ are calculated for the equilibrium (in-plane) orientation of $\mathbf{m}$, $\phi_0(\mathbf{B}, T)$, $\theta_0 = \pi/2$, and $\alpha_0$ is the Gilbert damping parameter. At the in-plane equilibrium orientation of $\mathbf{m}$, $F_{\theta\phi} = F_{\theta K_1} = F_{\theta K_u} = 0$ and there is no torque due to the thermal change of $M_0$.

In Eq. (4) the precession amplitude and phase are determined as $A_n(B) = \sqrt{a_n^2(B) + b_n^2(B)}$ and $\varphi_n(B) = \arctan(a_n(B)/b_n(B))$, where $a_n(B) = \left(-\omega_n \xi Q_n^b + 2\tau_M^{-1}\omega_n^2 Q_n^a\right)/G$ and $b_n(B) = \left(\omega_n \xi Q_n^a + 2\tau_M^{-1}\omega_n^2 Q_n^b\right)/G$, with $G = \xi^2 + 4\omega_n^2\tau_M^{-2}$, and $\xi = \omega_n^2 - \omega^2$. Here $\omega = 2\pi f(B)$ and $\tau_M^{-1} = \alpha_0 \gamma_0 (F_{\theta\theta} + F_{\phi\phi})/2$ are the field-dependent frequency and decay time of the fundamental magnon mode[12]. The Fourier series expansion coefficients of the effective driving force $Q_n^a$ and $Q_n^b$ are determined by the temporal modulation of the temperature given by Eq. (2) and are equal to $Q_n^a = \frac{2}{\pi}\frac{\omega_1 \tau_0}{1+\omega_n^2\tau_0^2}Q$ and $Q_n^b = \frac{2}{\pi}\frac{\omega_1 \omega_n \tau_0^2}{1+\omega_n^2\tau_0^2}Q$, where $Q = \gamma_0(\sin(4\phi_0)\beta_{K_1}/2 - \cos(2\phi_0)\beta_{K_u})\delta T$.

## Data availability
Raw pump–probe data (presented in Figs. 1 and 2) and processed data (presented in Fig. 3) that support the findings of this study are available in Mendeley Depository with the identifier https://doi.org/10.17632/h2yw7v8kb2.1

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

## Acknowledgements

We are grateful to Ilya Razdolski for fruitful discussions. The work was supported by the Bundesministerium fur Bildung und Forschung through the project VIP+ "Nano-magnetron," the Deutsche Forschungsgemeinschaft and the Russian Foundation for Basic Research (Grant No. 19-52-12065) in the frame of the International Collaborative Research Center TRR 160 and by the Engineering and Physical Sciences Research Council (Grant No. EP/H003487/1). The cooperation between TU Dortmund, the Lashkaryov Institute, and the Ioffe Institute was supported by the Volkswagen Foundation (Grant No. 97758).

## Author contributions

M.K. constructed the experimental setup, carried out the experiment, and performed the data analysis. A.V.S., I.A.A., and A.V.A. designed the experiment and supervised the experiment and theoretical modeling, T.L.L., S.M.K., and V.E.G. performed theoretical analysis and numerical modeling, A.W.R. and D.P.P. designed and deposited the Galfenol samples, D.P.P. characterized the Galfenol samples, M.K., A.V.S., T.L.L., S.M.K., V. E.G., I.A.A., A.W.R., A.V.A., and M.B. discussed the results and wrote the manuscript.

## Competing interests

The authors declare no competing interests.
