## [Peer Review File · Nature Communications]

Reviewers' comments:

Reviewer #1 (Remarks to the Author):

This manuscript describes an interesting concept and experiment on resonantly harvesting thermal energies (temperature oscillations) by using the magnon mode excitations. Experimental verification for the magnon excitation is demonstrated via the time-resolved Kerr effect. Data interpretation is given by using the calculated temperature variation in the context of the micromagnetic LLG theorem. The result is overall interesting and may present a broad impact on the field of ultrafast dynamics and spintronics. The work may be ultimately considered publishing in Nature Communications. However, there are some unconvincing aspects related to the experiment. The theoretical description is also quite preliminary. These aspects should be improved before the manuscript can be accepted.

My comments to the authors are:

(1) The Temperature calibration is rather unsatisfactory. It is basically estimated from a calculation without any additional validation. As far as I understand, this experiment was performed at room temperature. Was the experiment performed in a vacuum chamber or just ambient? The authors claim a global temperature rise of $\sim 10 - 42$ K and an oscillation of ~ 1 K. This sounds rather challenging to control and also to be able to excite coherent magnons. The authors should discuss the stability and temperature background. The values for the temperature estimates should also have errorbars with them.

(2) As far as I know, the experiment is performed at a single frequency, making it less convincing. What technical limitation prevents the author to try other frequencies? If not, I suggest the authors scan a small range of frequency for their laser heating drive (and subsequently detect Kerr dynamics and then FFT). This is probably not critical for the thermal part, but will be very helpful in understanding the magnon part. To validate that the spin dynamics are coherent, a dispersion (e.g. Kittel) relation should be obtained, including discussion on the linear and nonlinear regimes based on the power dependence.

(3) The authors claim that their technique can work in the GHz and even sub-THz regime. They should provide a more clearly bandwidth with good justification. This will involve the discussions of their setup, materials systems, and other potential limiting factors against extending the bandwidth both ends (lower GHz or upper THz).

(4) It seems that the underlying mechanism for such temperature-driven spin dynamics lies in the strong modulation effect of the magnetic anisotropy, which behaves like an effective field. The authors should add quantitative discussions on the magnitude and effectiveness of such modulation, using their material therein. In addition, is FeGa alloy chosen for some unique material concerns? Can other ferromagnets work as good as FeGa? If not, why.

(5) In addition to the temperature-induced anisotropy modulation, thermal gradient torques (via spin-Seebeck effect for example), if excited, may also contribute to the magnon excitation. This is a different mechanism where true spin torques are actually generated owing to the spin-orbit coupling (of FeGa, for example). There seem no relevant discussions on ruling out such in the present manuscript.

Reviewer #2 (Remarks to the Author):

This paper experimentally studied the magnon generation using pulsed laser heating on a $\text{Fe}_{0.81}\text{Ga}_{0.19}$ film grown on a GaAs substrate. The authors used a pump-probe method, where a periodic laser pulse is used to heat up the sample at a frequency of 10 GHz and used another laser at 1 GHz to probe the Kerr rotation signal to infer the magnon information. They then used a phenomenological model to link the magnetization precession amplitude to temperature variation excited by the pump laser. The paper is written in a way that is very difficult to understand and may appeal only to domain experts instead of a broader audience, which *Nat. Comm.* emphasizes. This is my feeling, but I am not using it as a criterion to judge the quality of the paper. It is the job of the editor.

Technically, I think the results are very solid and it is an interesting observation, but the physics is not clearly elucidated. Below are my major comments:

1. Most of the results show tuning the B field can lead to peaks in the Kerr rotation signal. If the B field is with a value that is off-resonance, there is little signal, but if it is on-resonance, there are peaks. I am almost lost on why you need a pulsed heating source at a specific frequency. What if you match your frequency to 20 GHz, which seems to have the largest amplitude in Fig. 1a? I actually only see a peak at 20 GHz there. To criticize lightly, the paper does not explain the rationales well.
2. I get that the results show heating can lead to larger magnetization precession amplitude, which presumably relates to the population of magnons. But how did this happen in a mechanistic level? Which phonon modes couple to which magnon modes? If electron is playing a role here? Seems that there is phonon-electron coupling in the picture as well. The phenomenological observation of heating leading to magnetization enhancement has been studied extensively. In my opinion, the use of a pulsed light is just another way of showing the same phenomena.
3. Magnon has a dispersion, meaning there is a spectrum of frequency with different wavevectors. How is this playing a role in the picture?
4. The energy conversion efficiency is extremely low. It is fine, but the authors started the paper with discussing energy conversion.

Reviewer #3 (Remarks to the Author):

This manuscript develops the idea of thermal energy harvesting by channeling it into magnon excitations in ferromagnetic metal thin films. The idea is novel and the paper deserves to get proper exposure through publication in *Nature Communications*. However, improvements are needed before it becomes suitable for publication. A major one is a discussion of (quantum) efficiencies in channeling resonant energy through modulated optical heat injection as compared to microwaves, a method that has been well known for decades. In other words, under what conditions can the authors imagine resonant optical heating having advantages and possibly becoming a good candidate for device applications?

A clarification is needed with regard to the feasibility of exciting magnon resonances which are based on the Lifshitz-Gilbert equation (1) so that the resonance condition can be attained. Typically fs lasers come with fixed rep rates so the generated temperature field can match the natural magnon frequencies only under some tenability conditions. This should be made clear in the text. As a follow-up question, does thermal field resonant attainment have to excite magnon resonances only at the peak, or a sufficient energy transfer can occur at off-peak magnetic field values?

The occluded temperature field of Eq. (2) involves a stationary background DT. This will grow with successive pulsing of the laser as more thermal energy is pumped into the system, including the ferromagnetic thin film. The authors cite 42K background heating which sounds quite high. What (counter-efficiency) role does the growing background play in the operation of the magnon resonance system and how would performance improve with efficiency of removal of that heat? It was already mentioned that increasing DT shifts the resonance Bn field (and no doubt broadens the peak due to increased random magnetic motion). Are there interface conditions between thin film and surrounding space to be taken into account for optimization? Also, following resonant magnon motion (i.e. absorption of the thermal energy by the magnon system resulting in resonant motion), does there follow further conversion to heat, likely at the surfaces of the thin layer where collective magnon motion may be degraded due to their spatial discontinuity? This is an issue of adiabaticity, I reckon. The authors anticipate this energy transfer concept "to generate magnons in ferromagnetic layers deposited on processors and other microchips operating at GHz and sub-THz clock frequencies." What lossy mechanisms can they propose that might compromise the efficiency of resonant amplitude through further thermal dissipation? Wouldn't additional dissipation terms have to be added to Eq. (1) to predict the inevitable further heat diffusion losses?

Minor issues: In Fig. 3b the 15-fold increase of AB is claimed against the baseline, horizontal dashed line, which, however, looks artificially drawn and not based on data. It can be noted that small variations in the exact position of the baseline can change the 15-fold figure drastically: for example, a slight rigid upward translation to meet the tail end of the peaked distribution would reduce that figure anywhere down to between 8-fold (left boundary) and 4-fold (right boundary).
On line 110 does KR signal stand for Kerr? On line 136, t_0 should be changed to τ_0 . Please update.

We are grateful to all reviewers for their useful comments and questions which helped us to strengthen our case. In particular, we have performed additional experiments on a thicker film, revised the manuscript accordingly and added four Supplementary Notes. Below we answer in detail the reviewer's questions and refer to the revisions made in the manuscript.

Reviewers' comments

Reviewer #1 (Remarks for the author).

This manuscript describes an interesting concept and experiment on resonantly harvesting thermal energies (temperature oscillations) by using the magnon mode excitations. Experimental verification for the magnon excitation is demonstrated via the time-resolved Kerr effect. Data interpretation is given by using the calculated temperature variation in the context of the micromagnetic LLG theorem. The result is overall interesting and may present a broad impact on the field of ultrafast dynamics and spintronics. The work may be ultimately considered publishing in Nature Communications. However, there are some unconvincing aspects related to the experiment. The theoretical description is also quite preliminary. These aspects should be improved before the manuscript can be accepted.

We are thankful to the reviewer for finding the results to be interesting. We accept the reviewer's comments about unconvincing aspects of the experiment and theoretical description. We have performed additional experiments to make the conclusions more convincing and extended the presentation of the theoretical part in the main text of the manuscript. We have added 4 Supplementary Notes, referenced in the main text accordingly.

Question 1.

The Temperature calibration is rather unsatisfactory. It is basically estimated from a calculation without any additional validation. As far as I understand, this experiment was performed at room temperature. Was the experiment performed in a vacuum chamber or just ambient? The authors claim a global temperature rise of $\sim 10 - 42$ K and an oscillation of ~ 1 K. This sounds rather challenging to control and also to be able to excite coherent magnons. The authors should discuss the stability and temperature background. The values for the temperature estimates should also have error bars with them.

The experiment was performed at ambient conditions at room temperature, which is now mentioned in the revised version. We have improved the theoretical analysis for estimation of the background temperature, ΔT . As experimental validation, we have obtained the values of ΔT from the shift of the resonance magnetic field [Fig. 3a] induced by the background heating and added corresponding discussion on page 5. The details of the calculations and validation procedure are also presented in the Supplementary Notes 1 and 2. The highest estimated value of ΔT remains well below the Curie temperature of Galfenol and, thus, background heating does not affect the amplitude of coherent magnons at the resonance conditions, which is determined by the differential temperature, δT . Its value is validated by comparison of the measured (symbols in Fig. 3d) and calculated values of the precession amplitude on page 7 of the revised manuscript. The error bars in Fig. 3b do not exceed much the size of the symbols and the small fluctuations from the linear dependence of the measured precession amplitude on power are clearly seen in Fig. 3d.

Concerning stability: We agree, that in a real device, the background temperature can vary with time due to variation of the used power, environment and heat sink. In the revised version we have estimated a change of the precession amplitude of 3% for a 20% change of the background temperature ΔT (page 7). The errors for the experimentally obtained ΔT are $\sim 10\%$ and given in the Supplementary Note 2.

Question 2

As far as I know, the experiment is performed at a single frequency, making it less convincing. What technical limitation prevents the author to try other frequencies? If not, I suggest the authors scan a small range of frequency for their laser heating drive (and subsequently detect Kerr dynamics and then FFT). This is probably not critical for the thermal part, but will be very helpful in understanding the magnon part. To validate that the spin dynamics are coherent, a dispersion (e.g. Kittel) relation should be obtained, including discussion on the linear and nonlinear regimes based on the power dependence.

(a) The 10 GHz laser is a unique laser system and the repetition rate cannot be tuned. In this respect our experiments are similar to the first microwave experiments on magnetic resonance, where the frequency of electromagnetic waves was fixed and all resonances were detected by tuning their frequencies by an external magnetic field. However, in our experiments the thermal excitation is not harmonic and, thus, includes several frequencies: the fundamental (10 GHz) as well as higher harmonics of it (20 and 30 GHz) (see the bottom panel of Fig. 1c (revised)). As a result, there are several resonant magnetic fields where the magnon frequency coincides with one of the excitation harmonics and the precession amplitude is drastically increased (Fig. 3a). No peaks in the field dependence of the precession amplitude are detected in the case of single pulse excitation, where the amplitude gradually decreases with increasing B .

(b). The spin dynamics are coherent in the sense that all magnons are driven by periodic thermal excitation with fixed, i.e. not random, phases. Harmonic oscillations are observed only at the resonance conditions, i.e. at the values of magnetic fields when $f=10, 20$ or 30 GHz (see $f(B)$ dependence in Fig. 1a), which also confirms that the precession phase is preserved for a long time. We have added a paragraph about the magnon **dispersion** on page 6 of the revised manuscript and give its detailed description in the Supplementary Note 4. There we show that the main contribution to the measured signals is given by the fundamental magnon mode (with zero wave vector). To illustrate the destructive role of magnons with finite wave vectors, we have examined the effects in a thick (105-nm) Galfenol layer, in which the broadened magnon spectrum consists of spectrally close lying higher-order magnon modes. This leads to fast dephasing of the thermally driven magnetization precession and decreases the efficiency of resonant energy transfer. The results for the thick layer are presented in the Supplementary Note 4.

(c) **Nonlinear effects**, which could accompany the high-amplitude magnetization precession, are not observed in our study. The resonant signals remain harmonic and their amplitudes depend linearly on W in the whole range of excitation powers. However, the shift of the resonance frequency with the increase of ΔT affects the linear dependence of the precession amplitude on power **at fixed magnetic field**. This temperature-induced “nonlinearity” is known since the conventional microwave experiments on ferromagnetic resonance [Phys. Stat. Sol. 8, K89 (1965)]. A corresponding statement about nonlinearity has been added to the manuscript on page 7.

Question 3.

The authors claim that their technique can work in the GHz and even sub-THz regime. They should provide a more clearly bandwidth with good justification. This will involve the discussions of their setup, materials systems, and other potential limiting factors against extending the bandwidth both ends (lower GHz or upper THz).

We have added a discussion about bandwidths on page 8. To reach *high, sub-THz frequencies*, the corresponding bandwidths should be available for both temperature and magnons.

a) Temperature modulation bandwidth. Here we refer to the picosecond acoustic experiment in Ref. [4] where the temperature in a 12nm Al film was modulated at frequencies up to 400 GHz. The thermal properties of ferromagnetic metals do not differ much from normal metals and thus such a modulation is

possible in thin films. Theoretically, for fixed parameters the amplitude of the temperature oscillations is proportional to the power and inversely proportional to the frequency up to sub-THz frequencies.

b) Magnon bandwidth. In Ref. [12] it is shown that in thin (Fe,Ga) films magnons with frequencies up to 100 GHz may be excited at the magnetic field of 3T. Moreover, novel metallic ferrimagnetic materials (e.g. Mn-Te compounds) possess narrow magnon resonances at frequencies up to hundreds of GHz at low magnetic fields [29].

c) From the *lower end*, there is no limit for temperature modulation while for magnons the lower frequency varies from hundreds of MHz (e.g. for garnets) up to several GHz in metallic ferromagnets.

Question 4.

It seems that the underlying mechanism for such temperature-driven spin dynamics lies in the strong modulation effect of the magnetic anisotropy, which behaves like an effective field. The authors should add quantitative discussions on the magnitude and effectiveness of such modulation, using their material therein.

The referee is absolutely right that the underlying mechanism is modulation of the magnetic anisotropy. This analysis is described in details in Ref. [14] for the parameters of the (Fe,Ga) used in the present work. In the revised version, we have extended the discussion part of the main text on page 6 adding Eq.2 for the free energy and of the Method section, where additional Eq.5 and more details necessary for the analysis are presented.

In addition, is FeGa alloy chosen for some unique material concerns? Can other ferromagnets work as good as FeGa? If not, why.

Yes, other ferromagnetic and ferrimagnetic materials will work as well. The critical properties for the suggested concept are the pronounced magneto-crystalline anisotropy and the long lifetime of the magnetization precession. Galfenol is an actively studied novel ferromagnetic material, which has these properties. It is a good testbed for demonstration of the general concept to transfer parasitic heat modulated at high frequency to magnons. In the revised version we have added a paragraph about other materials on page 8.

Question 5

In addition to the temperature-induced anisotropy modulation, thermal gradient torques (via spin-Seebeck effect for example), if excited, may also contribute to the magnon excitation. This is a different mechanism where true spin torques are actually generated owing to the spin-orbit coupling (of FeGa, for example). There seem no relevant discussions on ruling out such in the present manuscript.

Indeed, the diffusion of hot carriers from the excited area as well as the thermal gradient inside a ferromagnetic film (we estimate 4% temperature difference across the film) can induce ac- spin transfer modulated at the laser repetition rate, f_0 [15,16]. However, due to the uniform magnetization in a single thin ferromagnetic layer, this transfer does not produce a torque on the magnetization [17]. The dominant role of the thermal modulation of the magneto-crystalline anisotropy is also confirmed by the dependence of the precession amplitude excited by a single laser pulse on the direction of the external magnetic field. This dependence demonstrates a four-fold in-plane symmetry with a slight uniaxial distortion, which corresponds to the magneto-crystalline anisotropy of the studied layer. The corresponding discussion is given on page 5 of the revised manuscript.

Reviewer #2 (Remarks to the Author):

This paper experimentally studied the magnon generation using pulsed laser heating on a Fe_{0.81}Ga_{0.19} film grown on a GaAs substrate. The authors used a pump-probe method, where a periodic laser pulse is used to heat up the sample at a frequency of 10 GHz and used another laser at 1 GHz to probe the Kerr rotation signal to infer the magnon information. They then used a phenomenological link to connect the magnetization precession amplitude to temperature variation excited by the pump laser. The paper is written in a way that is very difficult to understand and may appeal only to domain experts instead of a broader audience, which Nat. Comm. emphasizes. This is my feeling, but I am not using it as a criterion to judge the quality of the paper. It is the job of the editor.

We are thankful to the referee for this general comment. We have revised the paper answering all questions of the referees and added four Supplementary Notes. We hope that the revised manuscript becomes suitable for the broader audience which Nature Communications addresses.

Question 1.

Technically, I think the results are very solid and it is an interesting observation, but the physics is not clearly elucidated. Below are my major comments:

Most of the results show tuning the B field can lead to peaks in the Kerr rotation signal. If the B field is with a value that is off-resonance, there is little signal, but if it is on-resonance, there are peaks. I am almost lost on why you need a pulsed heating source at a specific frequency. What if you match your frequency to 20 GHz, which seems to have the largest amplitude in Fig. 1a? I actually only see a peak at 20 GHz there. To criticize lightly, the paper does not explain the rationales well.

In general, the excitation should be periodic, but not necessarily pulsed. To explain the observed effect in a more clear way we have extended the description of the analysis adding the equation for the free energy (Eq.2) and two coupled equations (Eq.5) in the Method Section, which describe the excitation of magnetic resonance. In the text on page 3, we compare the physics of magnon excitation with the excitation of a simple one-dimensional oscillator with eigenfrequency f (i.e. the analog of the fundamental magnon frequency) by an external oscillating force. The version of the Landau-Lifshitz-Gilbert equation Eq. (1) in the main text of the manuscript describes the same physics but in a more complicated system due to the precessional origin of the magnetization vector \mathbf{m} in three-dimensional space. Our thermal excitation is a periodic but not harmonic force (see the spectrum in Fig. 1c (revised)). Its spectrum includes the fundamental frequency ($f_1=10$ GHz) and higher harmonics ($f_2=20$ GHz, $f_3=30$ GHz etc). The sum of all these harmonics forms an external oscillating force acting on the oscillator. Obviously, the oscillator driven by such a force has the same frequency components, but the amplitude of each harmonic depends on the detuning between the oscillator eigen-frequency, f , and the excitation harmonic f_i . In our experiment, the oscillator (magnon) frequency, f , can be tuned by the external magnetic field. To emphasize this, we have added to the revised Fig. 1a several other spectral lines, which characterize the magnon spectrum at several values of applied external magnetic field. When the frequency f coincides with one of the exciting harmonics, the amplitude strongly increases, and this effect is clearly seen in our experiment. In the off-resonance regime the magnon precession is not harmonic (see right panel of Fig.2) and the amplitude is much smaller than in the resonance case. The analogy with the excitation of a simple one-dimensional oscillator is valid for the magnons in our thin (Fe,Ga) film because there is only a single relevant magnon mode, the frequency of which depends on B as shown in Figs. 1a and 1b. We have added the Supplementary Note 4 where we present results for a thicker (105-nm) (Fe,Ga) film where the magnon spectrum is broad and no resonant features are detected.

Question 2.

I get that the results show heating can lead to larger magnetization precession amplitude, which presumably relates to the population of magnons. But how did this happen in a mechanistic level? Which phonon modes couple to which magnon modes? If electron is playing a role here? Seems that there is phonon-electron coupling in the picture as well. The phenomenological observation of heating lead to magnetization enhancement has been studied extensively. In my opinion, the use of a pulsed light is just another way of showing the same phenomena.

Indeed, heating of a ferromagnetic layer increases the thermal population. However, this is true for non-coherent magnons (with random phases) and this effect can be used for generation of spin currents (see, for instance, [Nature Mater. **11**, 391(2012)], for details). In our experiments the thermally excited magnetization precession is coherent and thus the phases of all excitations (thermal and magnons) are fixed and not random. In general, the excitation should be periodic, but not necessarily pulsed. For continuous excitation, the magnon phase is random and no resonances will be observed.

In our experiment, the **role of electrons** is important at the first stage of the optical excitation. The optical excitation heats the electrons and they transfer the energy to the lattice during a time of ~ 1 ps. Due to this short time, the temperature is modulated in time and the magnetization precession, the period of which is ~ 100 ps, is coherent. In the revised version this is mentioned on page 8.

Question 3.

Magnon has a dispersion, meaning there is a spectrum of frequency with different wavevectors. How is this playing a role in the picture?

The excitation spot is ~ 17 μm in diameter and only magnons with $q < 10^6$ m^{-1} are excited coherently. In a film, where the thickness h is much less than the excitation spot and $hq \ll 1$, the magnon dispersion in this range of q is negligible and lies within the spectral width of the fundamental magnon mode. The higher-order exchange magnon modes quantized along the film normal have significantly higher frequencies (> 160 GHz), which are out of the spectral range of thermal excitation and, therefore, are not excited. Thus, all excited magnons have the frequency of the fundamental mode which is mentioned in the revised version on page 6. The analysis of the magnon spectrum and the contribution of magnons with finite wave vectors as observed in the experiment with a thick (Fe,Ga) film are presented in the Supplementary Note 4. Please, see also our reply to the Question 2 of Reviewer 1.

Question 4.

The energy conversion efficiency is extremely low. It is fine, but the authors started the paper with discussing energy conversion.

We agree with the reviewer about the confusing start of the paper. We took out the first sentence of the abstract. We would like to mention that energy harvesting is aiming not only at realizing efficient cooling but also at transferring parasitic heat to other kinds of energies, i.e. coherent magnons in our case, which may be used for other purposes.

Reviewer #3 (Remarks to the Author):

This manuscript develops the idea of thermal energy harvesting by channeling it into magnon excitations in ferromagnetic metal thin films. The idea is novel and the paper deserves to get proper exposure through publication in Nature Communications. However, improvements are needed before it becomes suitable for publication.

We are thankful to the reviewer for studying the manuscript and we agree that improvements were needed.

Question 1.

A major one is a discussion of (quantum) efficiencies in channeling resonant energy through modulated optical heat injection as compared to microwaves, a method that been well known for decades. In other words, under what conditions can the authors imagine resonant optical heating having advantages and possibly becoming a good candidate for device applications?

It is not easy to compare the efficiency of resonant thermal and microwave excitation of coherent magnons because both methods depend on many technical issues, e.g. the design of the microwave cavity, efficiency of the transmission line, efficiency of the heat sink etc. A direct comparison of the precession amplitudes achieved in our experiment and in the experiment with a conventional coplanar microwave waveguide [26] for the same power shows that the efficiency of the resonant thermal excitation is one-order of magnitude lower. However, the effect of thermal modulation can be expressed as a virtual microwave field acting on a ferromagnet. The amplitude of this field in our experiment is 10 – 100 μT , which is comparable with the microwave induction generated by a microwave cavity [26]. By increasing the thermal modulation amplitude, it is possible to achieve efficiencies comparable with planar microwave microstructures. In the revised version, we give a direct comparison of our concept with conventional microwave and acoustic methods on page 7. We also discuss a way to improve the dynamical thermal efficiency $\zeta = \delta T / \Delta T$, which determines the fraction of the useful oscillating temperature δT relative to the total parasitic heating ΔT . This discussion has been added to page 7 of the manuscript and in detail to the Supplementary Note 3.

Question 2.

A clarification is needed with regard to the feasibility of exciting magnon resonances which are based on the Lifshitz-Gilbert equation (1) so that the resonance condition can be attained. Typically fs lasers come with fixed rep rates so the generated temperature field can match the natural magnon frequencies only under some tenability conditions. This should be made clear in the text. As a follow-up question, does thermal field resonant attainment have to excite magnon resonances only at the peak, or a sufficient energy transfer can occur at off-peak magnetic field values?

The reviewer is right that sufficient energy is transferred to the magnons only at the resonance condition which is achieved by tuning the external magnetic field. The width of the feature in Fig. 3b answers this question directly for the case of pulsed optical excitation. In the revised manuscript, the description of resonant conditions is given in more details, both on a qualitative and quantitative level, in the main text of the manuscript (pages 3 and 6), and Supplementary Note 4. It is important that these conditions are tolerant to variations of the background temperature ΔT which is mentioned in the revised manuscript on page 6. Please, see also the answer to Question 1 of Reviewer 1.

Question 3.

The occluded temperature field of Eq. (2) involves a stationary background ΔT . This will grow with successive pulsing of the laser as more thermal energy is pumped into the system, including the ferromagnetic thin film. The authors cite 42K background heating which sounds quite high. What (counter-efficiency) role does the growing background play in the operation of the magnon resonance system and how would performance improve with efficiency of removal of that heat? It was already mentioned that increasing ΔT shifts the resonance B_n field (and no doubt broadens the peak due to increased random magnetic motion).

Are there interface conditions between thin film and surrounding space to be taken into account for optimization?

Also, following resonant magnon motion (i.e. absorption of the thermal energy by the magnon system resulting in resonant motion), does there follow further conversion to heat, likely at the surfaces of the thin layer where collective magnon motion may be degraded due to their spatial discontinuity? This is an issue of adiabaticity, I reckon.

The authors anticipate this energy transfer concept “to generate magnons in ferromagnetic layers deposited on processors and other microchips operating at GHz and sub-THz clock frequencies.” What lossy mechanisms can they propose that might compromise the efficiency of resonant amplitude through further thermal dissipation?

Wouldn't additional dissipation terms have to be added to Eq. (1) to predict the inevitable further heat diffusion losses?

Before answering the Reviewer's comments, we would like to mention that the heat transferred to the magnons is very small relative to the heat generated as a result of optical excitation. The main idea of our work is to explore this parasitic heat for resonant excitation of magnons and we successfully show that the efficiency of heat transfer is high enough to exploit the generated coherent magnons in practice. In the comments, the Reviewer is raising questions concerning the decay of coherent magnons and the emission of phonons during this decay. We agree that these processes could play an important role in the magnon dynamics and we give the detailed answers below.

(a) The reviewer is right that the temperature background grows with successive pulsing. This heating ultimately reaches a stationary value ΔT . This value is still much lower than the Curie temperature and does not change significantly properties such as the decay of the magnetization. This is supported by the experimental fact that the amplitude depends linearly on the excitation power at the resonance conditions (see Fig. 3d).

In the revised version, the details of the calculation of ΔT are presented in the Supplementary Note 2. We have validated the calculated values for ΔT using the experimentally measured resonance shifts mentioned by the Reviewer and get reasonable agreement with the calculation. We introduce the dynamical resonant efficiency $\zeta = \delta T / \Delta T$, which gives the fraction of the oscillating temperature δT relative to the background heating ΔT . We discuss the dependence of $\delta T / \Delta T$ on various parameters and the design of optical and electrical devices. These details are presented in the Supplementary Note 3.

(b) The reviewer is right that heat diffusion losses of the magnons are, in general, inevitable. These losses are driven by the coherent field gradients and usually are growing with increasing average sample temperature. However, our experimental observations have evidenced the linear dependence of the coherent magnon amplitude on the pump laser power and the average temperature in the ranges realized in our experiments. This observation (equivalent to the observation of a temperature-independent width of the coherent magnon spectral line) indicates that the attenuation of the coherent magnons in our experiments is temperature independent.

(c). The reviewer is also right when suggesting that the highest gradients of the coherent ac- magnetic field generated by the fundamental, i.e. spatially homogeneous inside the film, magnetic mode are expected at the interfaces of the magnetic film with the capping metal layer and the semiconductor substrate. Our estimates demonstrate that the regime of the coherent oscillation in the film is isothermal rather than adiabatic. The wavelength of the thermal wave at the considered frequencies of 10 GHz - 30 GHz is longer than the total thickness of the magnetic film and of the metallic capping layer, and, thus, it exceeds the scale of potential spatial inhomogeneities that could be associated with diffusional transport at the interfaces. Thus, the experimentally observed coherent motion is isothermal.

(d). As the losses related to the diffusion processes are experimentally proved to be negligible, no additional temperature-dependent dissipation terms have to be added in Eq. (2) and a removal of the heat from the substrate, as suggested by the Reviewer, would not decrease the efficiency of the resonant amplitude excitation in the experimentally covered range of temperatures (around room temperature).

To summarize, we have demonstrated an original concept using the concept of a rather simple structure with an elementary design. We do not find anything that can be considered as a potential compromising factor or road blocker for application of our concept. Moreover, there are various perspectives for optimizing the suggested approach in detail that may need to be tailored for a specific application.

Minor issues.

In Fig. 3b the 15-fold increase of AB is claimed against the baseline, horizontal dashed line, which, however, looks artificially drawn and not based on data. It can be noted that small variations in the exact position of the baseline can change the 15-fold figure drastically: for example, a slight rigid upward translation to meet the tail end of the peaked distribution would reduce that figure anywhere down to between 8-fold (left boundary) and 4-fold (right boundary).

To be more precise, in the revised version of the manuscript the dashed line corresponds to the out-of-resonance background obtained by fitting it with the sum of Lorentzian peaks. The corresponding resonance increase in Fig.3b is 12-fold.

On line 110 does KR signal stand for Kerr? On line 136, t_0 should be changed to τ_0 Please update.

The corresponding changes were made. In the revised version we use “Kerr rotation signals” or “KR signals” through the text.

REVIEWERS' COMMENTS:

Reviewer #1 (Remarks to the Author):

The authors revised the manuscript and addressed the critics according to my earlier comments. I do not have additional comments

Reviewer #2 (Remarks to the Author):

I am satisfied with the revision and the authors' responses to my comments. I would support acceptance. -- Tengfei Luo

Reviewer #3 (Remarks to the Author):

The authors have responded in detail to my various comments, revised the main body of the manuscript and added to supplementary materials to improve the scientific approach to the subject matter. I find these improvements adequate and have no further comments or concerns with publishing the revised version.

Response to Referees:

Reviewer #1 (Remarks to the Author):

The authors revised the manuscript and addressed the critics according to my earlier comments. I do not have additional comments

Reviewer #2 (Remarks to the Author):

I am satisfied with the revision and the authors' responses to my comments. I would support acceptance.
-- Tengfei Luo

Reviewer #3 (Remarks to the Author):

The authors have responded in detail to my various comments, revised the main body of the manuscript and added to supplementary materials to improve the scientific approach to the subject matter. I find these improvements adequate and have no further comments or concerns with publishing the revised version.

We would like to thank all Reviewers for their valuable comments and suggestions.
We are happy we could improve our work in accordance to their remarks.